# Emergence of changing Central-Pacific and Eastern-Pacific El Niño-Southern Oscillation in a warming climate

Tao Geng [1,2], Wenju Cai [1,2,3] ✉, Lixin Wu [1,2] ✉, Agus Santoso [3,4,5], Guojian Wang [1,2,3], Zhao Jing [1,2], Bolan Gan [1,2], Yun Yang [6], Shujun Li[1,2], Shengpeng Wang[1,2], Zhaohui Chen [1,2] & Michael J. McPhaden [7]

El Niño-Southern Oscillation (ENSO) features strong warm events in the eastern equatorial Pacific (EP), or mild warm and strong cold events in the central Pacific (CP), with distinct impacts on global climates. Under transient greenhouse warming, models project increased sea surface temperature (SST) variability of both ENSO regimes, but the timing of emergence out of internal variability remains unknown for either regime. Here we find increased EP-ENSO SST variability emerging by around 2030 ± 6, more than a decade earlier than that of CP-ENSO, and approximately four decades earlier than that previously suggested without separating the two regimes. The earlier EP-ENSO emergence results from a stronger increase in EP-ENSO rainfall response, which boosts the signal of increased SST variability, and is enhanced by ENSO non-linear atmospheric feedback. Thus, increased ENSO SST variability under greenhouse warming is likely to emerge first in the eastern than central Pacific, and decades earlier than previously anticipated.

The El Niño-Southern Oscillation (ENSO) is the strongest and most consequential year-to-year climate fluctuation on the planet, with significant societal and environmental impacts that are felt worldwide[1–5]. Alternating between a warm El Niño phase and a cold La Niña phase, ENSO sea surface temperatures (SSTs) exhibit diverse anomaly centres[6–9], ranging from the equatorial eastern-Pacific (EP) to the equatorial central-Pacific (CP), referred to as EP-ENSO and CP-ENSO regimes, respectively[10,11]. The EP-ENSO regime is characterised by stronger warm-than-cold SST anomalies, whereas the CP-ENSO regime features larger cold-than-warm SST anomalies.

Strong EP-ENSO events, as seen in extreme El Niño in 1982 and 1997, caused a substantial disruption of marine ecosystems across the Pacific[2,12]. The large warm anomalies lead to a displacement of atmospheric deep convection towards the climatologically dry eastern equatorial Pacific, bringing floods to eastern-Pacific-rim countries but droughts in regions bordering the western Pacific[1,3]. CP-ENSO events, with its anomaly centre triggering an atmospheric response in the central-western equatorial Pacific, have different global impacts[7,13,14]. For example, catastrophic floods normally seen in parts of Ecuador and northern Peru during large EP El Niño are absent when CP El Niño occurs[15]. Different tropospheric and stratospheric responses are found in the middle to high latitudes with a strong interhemispheric difference and are projected to change in future climate[14]. Contrasting climatic impacts associated with the two ENSO regimes are also evident in North America[6], south Asia[16], Australia[17], as well as in North Atlantic tropical cyclones[18] and southern hemisphere storm-track activity[19], with differing implications for polar climate changes[20]. ENSO-related mitigation and

[1]Pilot National Laboratory for Marine Science and Technology (Qingdao), Qingdao, China. [2]Frontiers Science Center for Deep Ocean Multispheres and Earth System and Key Laboratory of Physical Oceanography, Ocean University of China, Qingdao, China. [3]Centre for Southern Hemisphere Oceans Research (CSHOR), CSIRO Oceans and Atmosphere, Hobart, TAS, Australia. [4]ARC Centre of Excellence for Climate Extremes, University of New South Wales, Sydney, NSW, Australia. [5]Climate Change Research Centre, University of New South Wales, Sydney, NSW, Australia. [6]College of Global Change and Earth System Science, Beijing Normal University, Beijing, China. [7]NOAA/Pacific Marine Environmental Laboratory, Seattle, WA, USA. ✉e-mail: wenju.cai@csiro.au; lxwu@ouc.edu.cn

adaption efforts, therefore, must consider the diversity in the response of each ENSO regime to greenhouse warming.

Given vast differences in their impacts, determining how ENSO may respond to greenhouse warming has been carried out by examining the response of EP-ENSO and CP-ENSO, respectively[5,21,22]. Although uncertainty exists[5], there is an emerging consensus on the increase in future ENSO SST variability by considering the response of the two types of ENSO separately[23,24]. The increase is underpinned by a warming-induced intensification of ocean-atmosphere coupling. The multi-model-based consensus, however, does not provide information on when the projected change in each ENSO regime emerges from its natural variations, or whether the two distinct ENSO regimes have the differing time of emergence (ToE). Such ToE information is critical for response strategies and timelines of policy setting[25].

Without considering ENSO diversity, a recent study focused on a comparison of ToE of ENSO-related rainfall variability with that of ENSO-related SST and found that ENSO-related rainfall change emerging some three decades earlier than that of ENSO-related SST, with ToE of ENSO SST variability occurring at ~2070 in about 70% of the latest climate models in localised region of the central-eastern equatorial Pacific[26]. Crucially, the ToE of the two ENSO regimes is unknown.

Like previous studies on ToE of mean climate changes[27-30] or climate extremes[31-33], there was an assumption of a linear relationship of a concurrent change in ENSO variability with rising global-mean temperature. For ENSO SST variability, however, its evolution under greenhouse warming may not be unidirectional[34], challenging the assumption. Here, without involving such an assumption, and by considering the ToE of each ENSO regime separately, we find that ToE of greenhouse warming-induced strengthening of EP-ENSO SST variability occurs earlier than that of CP-ENSO, and about four decades earlier than previously suggested.

## Results

### Two ENSO regimes and depiction of their ToE
As in the previous studies[10,23,35,36], the two ENSO regimes in observations and models have been obtained via an empirical orthogonal function (EOF) analysis of monthly SST anomalies in the tropical Pacific, yielding two principal patterns and their corresponding principal component (PC) time series (see "E and C indices" in Methods). The EP-ENSO and CP-ENSO are constructed through a linear combination of these two principal modes (Supplementary Fig. 1a, b). An E-index and a C-index, defined as $E = (PC1 - PC2)/\sqrt{2}$ and $C = (PC1 + PC2)/\sqrt{2}$, respectively, describe their temporal evolution[10,23,35].

We use outputs from 68 state-of-the-art climate models participating in the Coupled Model Intercomparison Project phase 5 (CMIP5)[37] and phase 6 (CMIP6)[38]. Each model consists of three experiments: a multi-century (more than 300 years) pre-industrial control simulation (piControl), a historical run starting from piControl in the mid-19th century and forced with historical forcings till 2005 for CMIP5 or 2014 for CMIP6, and a future warming experiment thereafter under the Representative Concentration Pathway (RCP) scenario for CMIP5 or the equivalent Shared Socioeconomic Pathway (SSP) scenario for CMIP6, with most models sharing a common integration period of 1850–2099 (Supplementary Table 1). SST anomalies of the three experiments with reference to the monthly climatology of piControl are detrended and concatenated to delineate ENSO evolution over the whole period spanning from piControl to future warming climates. We apply EOF analysis to the concatenated SST anomalies in each model to obtain the corresponding E-index and C-index, identifying the associated SST anomaly centres unique to each model. The multi-model mean (MMM) patterns reasonably resemble the observed patterns (Supplementary Fig. 1c, d).

The presence of the two ENSO regimes is underpinned by a non-linear Bjerknes feedback; specifically, zonal winds respond non-linearly to warm SST anomalies after large warm anomalies establish atmospheric deep convection in the equatorial eastern-Pacific, in turn, conducive to further SST warming by promoting the oceanic zonal advection, Ekman pumping, and thermocline feedback, constituting a positive feedback loop[35,39]. Consequently, positive SST anomalies in the equatorial eastern-Pacific grow far greater than cold anomalies. A strong El Niño in the equatorial eastern-Pacific leads to a large discharge of heat and a shallowed thermocline, conducive to strong La Niña in the central-Pacific, where cold anomalies are greater than warm anomalies. The degree of this ENSO non-linearity can be measured by the strength of a non-linear relationship[23,35,36] between PC1 and PC2, or by the sum of the magnitude of the two skewness values of E-index and C-index (Supplementary Fig. 1e, f; see "Observed and simulated ENSO non-linear dynamics" in Methods).

To determine ToE of simulated ENSO SST variability (i.e. standard deviation of SST time series) change, we adopt a commonly used "signal-to-noise" method[29,30] (see "ToE of ENSO SST variability change" in Methods), but without the assumption of a concurrent and linear relationship between ENSO SST variability change and rising global-mean temperature. Time series of EP- and CP-ENSO SST variability are first diagnosed using the E-index and C-index, respectively, in sliding windows of a given time length, moving every year (12 months) forward from the start of piControl to the end of the 21st century. Noise is defined as one standard deviation of the time series calculated over the entire piControl period, as any unforced internal variability is viewed as background "noise" to warming-induced ENSO SST variability change in the period from 1850 to the 21st century relative to the noise level of piControl, which we refer to as a signal. ToE is defined as when the signal first exceeds a threshold of two times of noise (i.e. Signal-to-Noise Ratio, SNR > 2.0; or the signal exceeds the 95-percentile threshold of the piControl variability range) and remains above the threshold thereafter. The ToE is determined for the E-index and C-index, separately.

The chosen threshold (2.0) reflects the statistical confidence level above 95% and is a reasonably conservative approach, as illustrated in an example model of GFDL-ESM4 for EP-ENSO (Fig. 1a, b) and CP-ENSO (Fig. 1c, d). Our approach differs from previous ToE methods, which involve fitting an evolution into a highly smoothed global-mean temperature series[26,29]. We record the ToE as the last year of the sliding window. Below we show that substantial differences exist in ToE between the two ENSO regimes.

### Earlier emergence of EP-ENSO change than CP-ENSO
We focus on the RCP85/SSP585 scenario and discuss other scenarios as sensitivity tests. Under greenhouse warming, there is a significant increase in the amplitude of both EP-ENSO and CP-ENSO[5,24], underscored by a strong inter-model agreement with 83.8% (57 out of the 68) models producing an increase in E-index variability, and 70.6% (48 out of 68) models producing an increase in C-index variability from piControl to 1900–2099 (Supplementary Fig. 2).

However, the increase in ENSO SST variability over a multi-centennial period does not guarantee its emergence before 2100, since the ToE of each ENSO regime depends on the length of time over which background SST noise and signal are diagnosed (Supplementary Fig. 3). For example, in the range of 30–50 years, only 57.14% and 29.41% of models show a ToE for E-index and C-index, respectively, whereas in the range of 60–80 years, 63.2% of models show a ToE of E-index and 33.8% of a ToE of C-index before 2100. For a given model, the longer the time window, the lower the noise level of unforced natural variability is compared to the warming-induced ENSO SST variability change signal, therefore a larger SNR (Supplementary Fig. 4), thus facilitating an earlier emergence of the signal and a higher probability of the emergence (Fig. 2a, b). Importantly, there is a consistently earlier ToE of increased E-index variability than that of C-index in all time lengths of diagnosis, supported by a larger portion of models that show a ToE before 2100 for E-index than that for

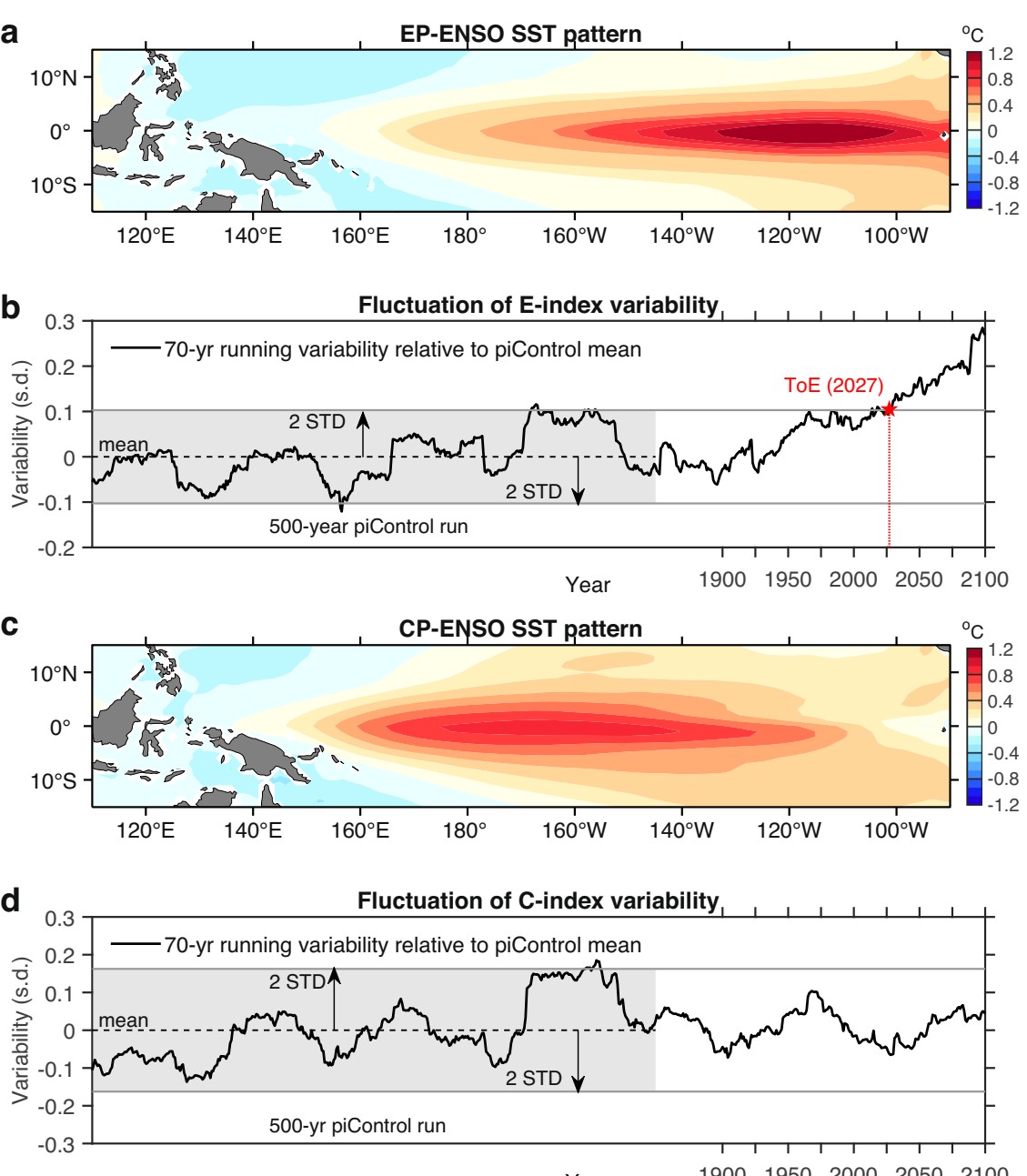

**Fig. 1 | Emergence of El Niño-Southern Oscillation (ENSO) sea surface temperature (SST) variability change from noise.** Shown is an illustrative example from GFDL-ESM4, a climate model able to realistically simulate non-linear ENSO characteristics (see Supplementary Fig. 1). **a** Spatial pattern of eastern-Pacific (EP) ENSO, obtained by regressing grid-point SST anomalies (°C) onto E-index over the whole period from piControl to 2100. **b** Evolution of E-index variability (black solid line; s.d.) relative to its mean pre-industrial level (dashed black line), in a 70-year sliding window moving forward by 1 year from the start of piControl and recorded at the last year of the time window. The red star marks the time of emergence (ToE) of E-index variability out of its internal variation bound, measured by two standard deviations (STD) of the 70-year running variability in piControl (grey shading), and remaining outside thereafter. **c, d** Same as **a, b**, respectively, but for C-index. Results shown here are based on the RCP85/SSP585 scenario for a future warming climate.

C-index. These features are seen in separate CMIP5 and CMIP6 ensembles.

The dependence of ToE on time length exhibits a saturation behaviour (Fig. 2a, b). For a time length longer than ~60–80 years, internal variability hardly decreases and the warming-induced signal is little increased with a longer time length. The behaviour indicates that a time window longer than 60–80 years would only marginally enhance the detection of ENSO variability change. Given that relatively reliable observations are available for ~70 years (observations before 1950 carry a large uncertainty[40]), below we focus on detectability for

the time length of 60 to 80 years to provide an assessment of ToE relevant to observed ENSO detection.

Within this 60–80-year range, 63.2% (43 out of 68) of all models (Fig. 2c), or 75.4% (43 out of 57) of models that produce an increase in E-index variability relative to piControl (Supplementary Fig. 2a), generate a ToE of EP-ENSO before 2100. These percentages, by contrast, decrease to 33.8% (23 out of 68) and 47.9% (23 out of 48) for CP-ENSO (Fig. 2d; Supplementary Fig. 2b).

For each model that shows a pre-2100 ToE of E-index, an averaged ToE of the 21 values in the 60–80-year window is calculated. A

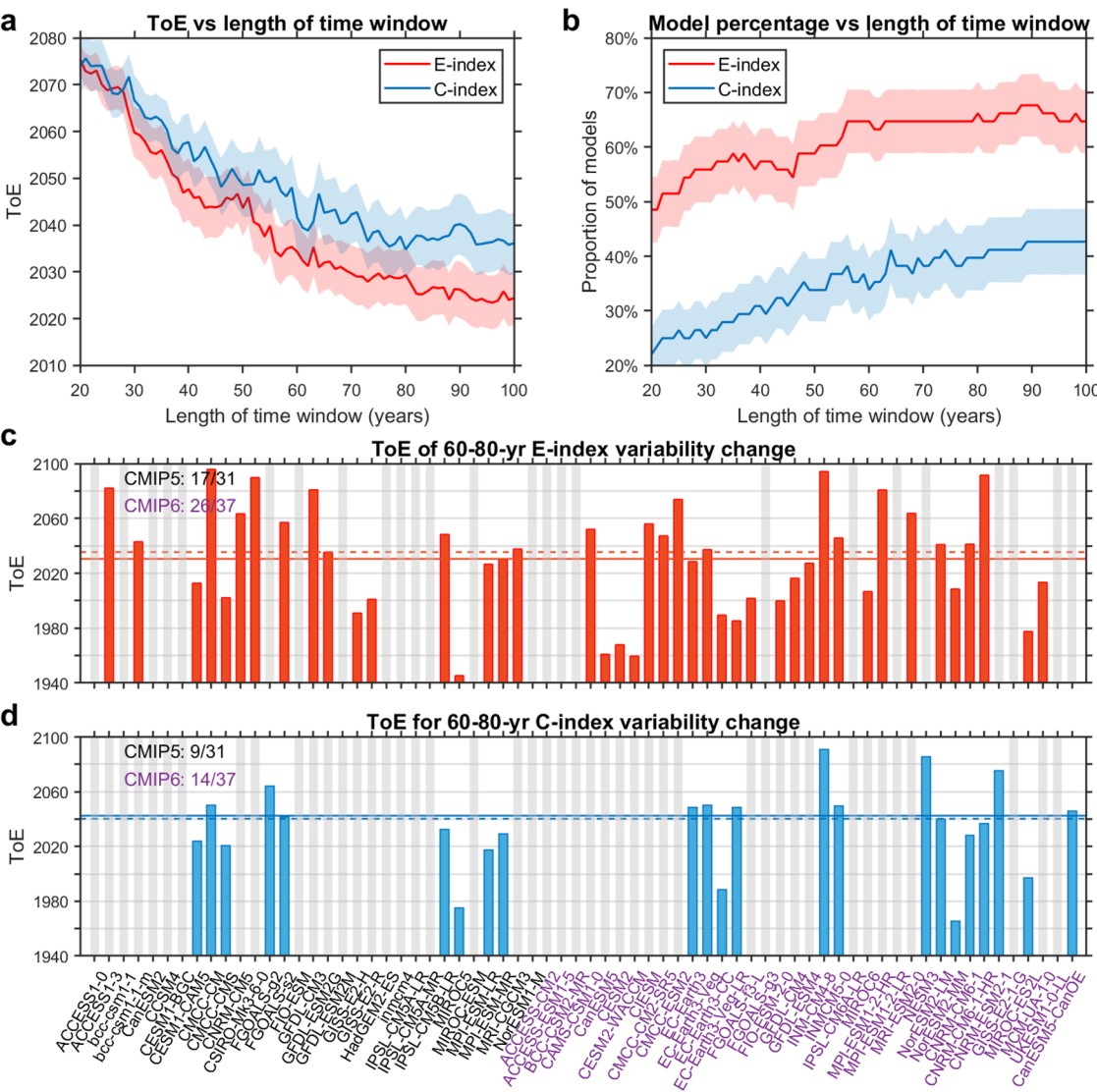

**Fig. 2 | Earlier time of emergence (ToE) of eastern-Pacific El Niño-Southern Oscillation (EP-ENSO) than central-Pacific (CP) ENSO. a** Dependence of ToE upon length of time window over which signal and noise are diagnosed for E-index (red) and C-index (blue). **b** As in **a**, but for proportion (in percentage) of models that show emergence of E-index (red) and C-index (blue) before 2100. Solid lines and shadings indicate multi-model mean and one standard deviation of a total of 10,000 inter-realisations, respectively, based on a Bootstrap method (see Methods). **c** ToE diagnosed from a 60–80-year sliding window for E-index variability in all 31 CMIP5 and 37 CMIP6 models. Only models that show ToE spanning the whole 60–80-year window are indicated by red bars, otherwise they are greyed out. Multi-model mean and median values are indicated by the horizontal solid and dashed lines, respectively. Numbers on the upper left denote the number of models showing ToE in the CMIP5 (black) or CMIP6 (purple) multi-model ensemble. **d** As in **c**, but for C-index. Results shown here are based on the RCP85/SSP585 scenario for a future warming climate.

MMM ToE value of 2030 is obtained (2035, if using median). However, inter-model differences are large, indicating that ToE based on a single realization of climate (as is the case for observations) can be uncertain. The uncertainty is estimated to be ~6 years for the multi-model mean based on a Bootstrap technique (see "Bootstrap test" in Methods). For CMIP6 only, ~70% (26 out of 37) models show a ToE before the end of the 21st century, yielding a mean ToE of 2026, implying that a stronger EP-ENSO would likely start to emerge above its pre-industrial level; by contrast, increased CP-ENSO variability emerges more than one decade later, with a mean ToE of 2043 in ~34% (23 out of 68) of all models, or 2040 in ~38% (14 out of 37) of CMIP6 models.

The analysis above indicates that the earlier ToE of increased EP-ENSO SST variability is supported by its higher probability of a ToE before 2100. These features are seen in all emission scenarios or using different threshold values of SNR to define the ToE (Supplementary Table 2). The ToE of EP-ENSO around 2030, a

decade or two earlier than CP-ENSO, appears in all emission scenarios and its probability increases with the intensity of greenhouse gas emissions.

Our result of increased EP-ENSO SST variability emerging at around 2030 is in stark contrast to a recent study comparing ToE of ENSO-related rainfall variability with that of ENSO-related SST variability and finding that ENSO-related SST variability emerges at about 2070 in a localised region of the central-eastern equatorial Pacific[26]. Their result is based on a multi-model average using CMIP6 models that show ToE in similarly 70% of the models. The difference stems from a different treatment of ENSO SST variability response. Without pre-assuming that ENSO SST evolution is concurrently and linearly varying with the rising global-mean temperature, our approach allows non-linear or non-unidirectional evolution of ENSO SST variability change under greenhouse warming[34]. In addition, noting that the real world has relatively reliable post-1950 SST data, we use a time length of 60–80 years to diagnose signal and noise, longer than the 30-year

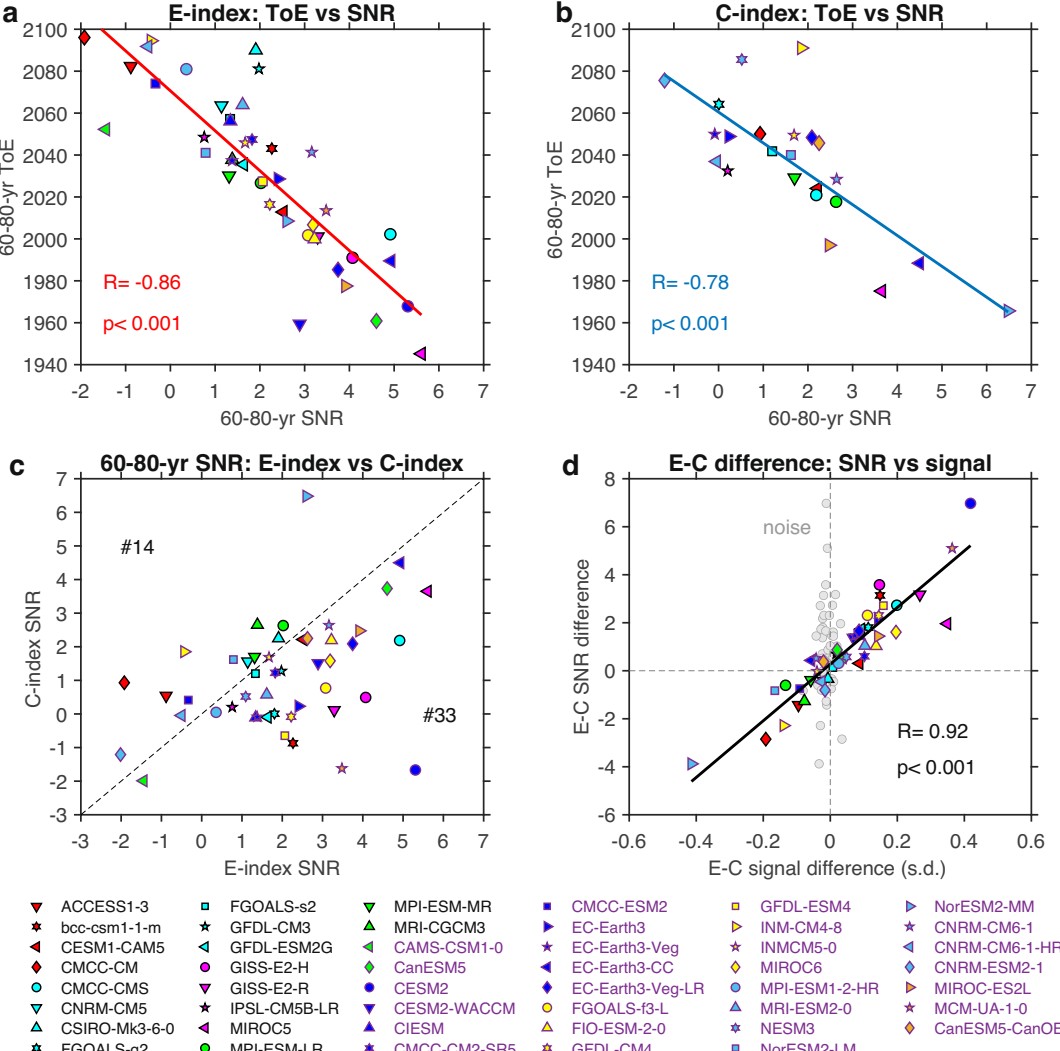

**Fig. 3 | El Niño-Southern Oscillation (ENSO) time of emergence (ToE) determined by magnitude of sea surface temperature (SST) variability change.**
**a**, **b** Inter-model relationship between ToE and Signal-to-Noise-Ratio (SNR) in models (symbols; name of CMIP5 models in black, name of CMIP6 models in purple) that show ToE before 2100 for **a** E-index and **b** C-index. **c** Contrast of SNR between E-index (*X*-axis) and C-index (*Y*-axis). Number on the top left, or bottom right, indicates the number of models showing larger SNR of C-index, or E-index, respectively. **d** Inter-model relationship between E-minus-C difference of signal and E-minus-C difference of SNR. For comparison, noise levels are shown in grey filled circles. Here we focus on the 60–80-year window. Signal is calculated as the 1960–2030 E-index or C-index variability change from the mean level of 60–80-year windowed variability in piControl, and noise is defined as one standard deviation (s.d.) of the 60–80-year windowed variability in piControl. Linear fits (solid lines) are displayed in **a**, **b**, **d** together with correlation coefficient R and corresponding *p*-value. Results shown here are based on the RCP85/SSP585 scenario for a future warming climate.

period they used. Further, we consider diversity in ENSO SST patterns that vary vastly across models, and separate ENSO SST response into orthogonal EP- and CP-ENSO regimes unique to each model, which allows the better manifestation of the signal[23]. The MMM ToE of EP-ENSO occurring around 2030 implies that the intensified EP-ENSO SST would likely emerge out of its pre-industrial level within the next decade or so. Below we examine the dynamics underpinning the earlier ToE of EP-ENSO change.

**Signal of increased SST variability determines ToE**
Given that ToE is determined by the extent to which the signal of SST variability change exceeds the level of noise, we compare the role of signal and noise in controlling the inter-model differences of ToE. For ease of comparison, the signal is hereafter calculated as the E-index or C-index variability in 1960–2030 relative to the mean level of 60–80-year windowed variability in piControl. Using signal averaged over the 1900–2100 period yields similar results. For both EP-ENSO and CP-ENSO, as expected, an earlier ToE is systematically associated with a

larger SNR (Fig. 3a, b). The model-dependent SNR is, in turn, determined by the magnitude of the signal of SST variability change more than by that of noise (Supplementary Fig. 5a, b), highlighting that an earlier ToE is principally attributed to a stronger ENSO response to climate change (Supplementary Fig. 5c, d).

In a similar vein, the earlier emergence of EP-ENSO relative to that of CP-ENSO is associated with a stronger signal of E-index variability change compared to that of C-index. In 47 models in which the emergence of either EP- or CP-ENSO occurs before 2100, more than 70% of models (33 out of 47) produce a larger SNR of E-index compared to that of C-index (Fig. 3c), consistent with the above result that a higher portion of models generates a ToE of EP-ENSO before 2100. The SNR difference between E-index and C-index is attributable to the difference in the signal, with an inter-model correlation reaching over 0.9 (Fig. 3d). As a result, the ToE difference between EP-ENSO and CP-ENSO is mainly caused by their different SST response to greenhouse warming rather than by their different natural variability.

Previous studies suggested that a warming-induced enhancement of ocean stratification featuring faster warming in the upper-ocean than that below contributes to an increase in EP-ENSO SST variability by boosting the dynamical coupling between ocean and atmosphere in a warming climate[23]. The increase in EP-ENSO variability is, in turn, conducive to an increase in CP-ENSO variability, as strong CP La Niña tends to occur after a large upper-ocean heat discharge induced by a previous strong EP El Niño[22,41,42]. Indeed, the signal of both EP-ENSO and CP-ENSO variability change is associated with an enhanced upper-ocean stratification in the equatorial Pacific (Supplementary Fig. 6a, b). However, an inter-model correlation between the E-minus-C signal contrast and the stratification change shows that changes in ocean stratification account for only ~16% (r = 0.39) of the E-minus-C difference in the SST variability change across models (Supplementary Fig. 6c), suggesting that other mechanisms are at work. Below we show differences in the response of rainfall variability associated with EP-ENSO and CP-ENSO are the catalyst for the E-minus-C signal contrast and the E-minus-C ToE contrast.

## Mechanisms controlling stronger EP-ENSO SST change signal

One of the most prominent differences in the response of ENSO to greenhouse warming is that the associated rainfall variability increases more in the EP region than in the CP region[43,44], even if SST warming is uniform across the equatorial Pacific[43]. However, most climate models project a long-term pattern featuring faster warming in the eastern equatorial Pacific than the surrounding oceans[5,22,45], further enhancing the contrast in rainfall variability between EP-ENSO and CP-ENSO. The pattern of SST warming, though different from the recently observed[5], is highly possible in the future as greenhouse effects progressively overwhelm other factors that, in observations, temporarily mask the warming signal, including decadal climate variability[46–48]. The warming pattern facilitates the establishment of atmospheric deep convection in the eastern equatorial Pacific, where non-linear Bjerknes positive feedback operates, promoting variability of rainfall, wind, and hence EP-ENSO SST, ultimately contributing to the E-minus-C signal contrast, as illustrated below.

We calculate local SST warming relative to the tropical average (20°S–20°N) from piControl to 1960–2030, scaled by the increase in global-mean SST in each model (to account for the difference in climate sensitivity). An inter-model regression shows that a stronger E-minus-C signal contrast in SST variability change is associated with faster warming in the eastern equatorial Pacific (Fig. 4a). Strikingly, the zonal warming differential, defined as SST warming difference between the eastern (170°W–100°W, 5°N–5°S) and western (120°E–170°W, 5°N–5°S) equatorial Pacific (dashed boxes in Fig. 4a), is systematically linked to the E-minus-C signal contrast, with a significant inter-model correlation of r = 0.7 (p < 0.001). Thus, the east-west warming differential accounts for ~50% (r²) of the E-minus-C signal contrast across models, much higher than that associated with the vertical stratification change.

The influence of zonal warming differential is exerted by altering atmospheric convection and the associated rainfall response. To illustrate, we regress rainfall anomalies onto E-index and C-index, respectively. The multi-model mean regression map of EP-ENSO rainfall anomalies shows a zonally elongated response towards the eastern equatorial Pacific, indicating an eastward extension of atmospheric convection; by contrast, the pattern for CP-ENSO rainfall anomalies is confined to the west of the dateline (shading in Supplementary Fig. 7a, b). Compared to piControl, the rainfall response to a unit of SST anomaly increases under greenhouse warming[49], signifying an intensified ocean-atmospheric coupling, more so for that associated with EP-ENSO (contours in Supplementary Fig. 7a, b). An inter-model regression shows that a stronger E-minus-C signal contrast is systematically associated with an increase in rainfall variability, which is seen to the east of 180° extending into the eastern equatorial Pacific

(shading in Supplementary Fig. 7c), coinciding with regions where total rainfall variability increases under greenhouse warming (contour in Supplementary Fig. 7c). The spatial overlap highlights that E-minus-C signal contrast is linked to the increase in convective rainfall variability toward the east. As expected, this linkage is also seen in the variability of zonal winds (Supplementary Fig. 8).

Because of large inter-model differences in ENSO SST anomaly centres, ENSO-related rainfall anomaly pattern or anomaly centre differs vastly from one model to another. To gauge rainfall variability change from that of piControl, we take full-period time series of rainfall anomalies at the anomaly centres unique to each model. The centre is defined as the location of the maximum regression coefficient of grid-point quadratically detrended rainfall anomalies onto the E-index or C-index in each model. We then normalise each time series of the detrended rainfall anomalies and calculate rainfall variability change (1960–2030 minus piControl) at each centre. The E-minus-C contrast in the rainfall variability change shows a statistically significant inter-model correlation with the east-minus-west SST warming differential (r = 0.76, p < 0.001) (Fig. 4b), reinforcing that the zonal warming differential is linked to the E-minus-C contrast in the rainfall variability change.

The faster warming in the eastern equatorial Pacific takes atmospheric convection toward the east, where the air-sea coupling is strengthened under greenhouse warming in part due to an eastward migration of the rising branch of the Walker circulation[50,51]. In a linear system, the associated strengthening in wind variability would, in turn, intensify EP-ENSO SST variability through ENSO positive feedback loop[52], enhancing its signal of SST variability change and the contrast to CP-ENSO. Models that simulate a larger E-minus-C contrast in the rainfall variability change systematically simulate a greater E-minus-C signal contrast in SST variability change (Fig. 4c).

The eastern equatorial Pacific is also the centre of the non-linear Bjerknes feedback, which is far weaker in the central and western Pacific[39,42], further enhancing the E-minus-C signal contrast. The strength of the non-linear Bjerknes feedback is reflected in the opposite-signed skewness of E-index and C-index diagnosed over the full concatenated period (Supplementary Table. 1), or the non-linear coefficient "α" (Alpha) in the quadratic relationship between PC1 and PC2 of monthly SST anomalies (Supplementary Fig. 1; see "Observed and simulated ENSO non-linear dynamics" in methods). There is a statistically significant inter-model correlation between Alpha and signal of E-index variability change or between Alpha and E-minus-C signal contrast in SST variability change (Supplementary Fig. 9), affecting their ToE. For example, seven out of ten models with the largest *Alpha* show a ToE of E-index before 2100, with a MMM ToE over the seven models in 2018, whereas only five out of ten models with the smallest *Alpha* produce a ToE of E-index before 2100, with a MMM ToE of 2044.

The non-linear Bjerknes feedback affects the E-minus-C signal contrast and the ToE contrast through two pathways. One is by inducing an exponentially growing zonal wind response to further SST warming after the establishment of atmospheric deep convection. As ENSO-related atmospheric convection migrates more frequently towards the eastern equatorial Pacific in a warming climate, the non-linear Bjerknes feedback is activated more readily, which further enhances rainfall and zonal wind variability east of the International Dateline (180°). This process is seen in an inter-model regression of rainfall variability change, or zonal wind variability change, onto the strength of the non-linear feedback measured by the amplitude of Alpha (Fig. 5a, b). Models that simulate stronger non-linear feedback produce a stronger increase. Another pathway for the impact is by non-linear rectification of the mean SST warming. A stronger ENSO non-linearity causes a stronger rectification of EP-ENSO change onto the mean state, reinforcing the zonal warming differential[53]. As such, models with a stronger non-linear Bjerknes feedback (i.e., a larger

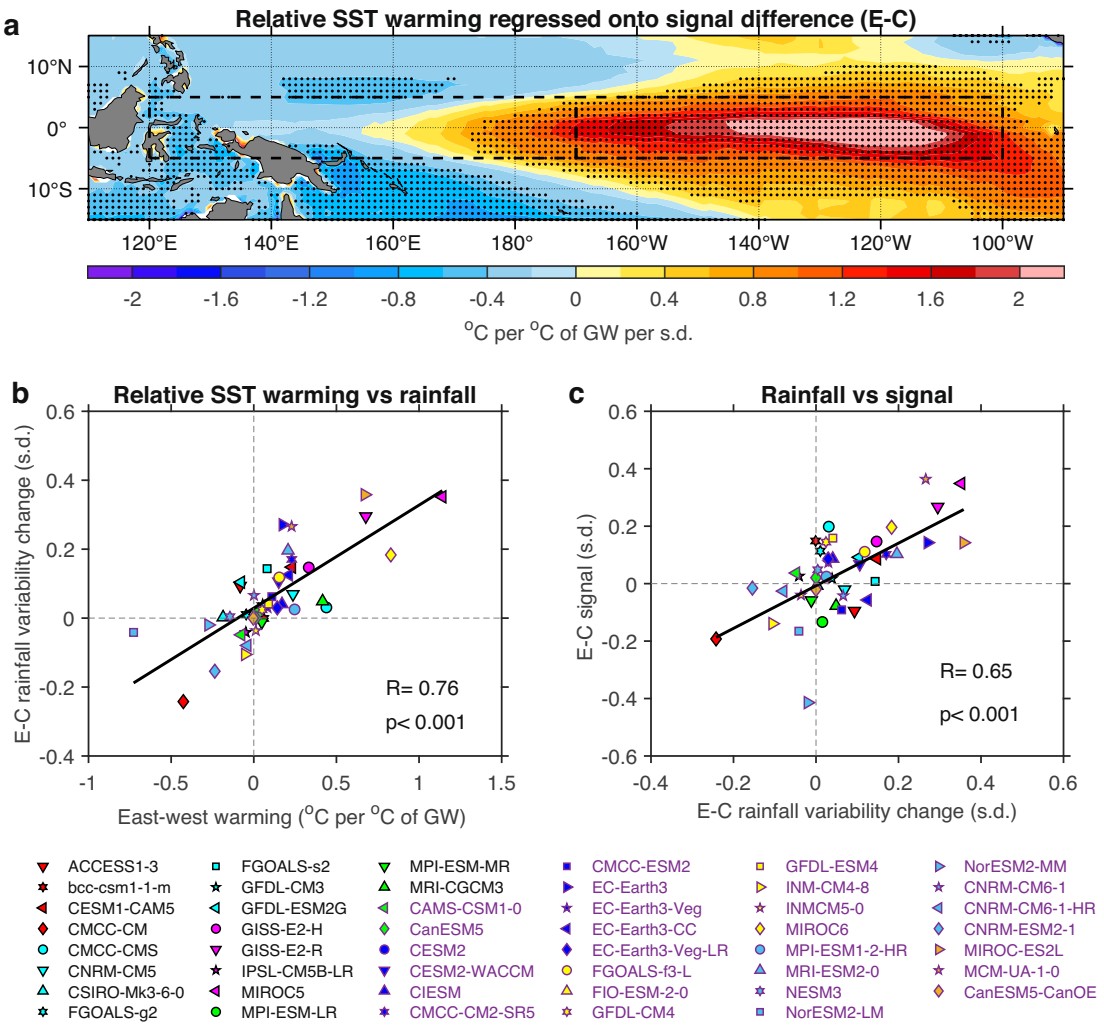

**Fig. 4 | Faster eastern-Pacific warming facilitates stronger signal of increased eastern-Pacific El Niño-Southern Oscillation (EP-ENSO) sea surface temperature (SST) variability. a** Inter-model regression of relative SST warming onto E-minus-C signal difference (s.d.). Relative warming is calculated as mean SST change (with 20°S–20°N average removed) between 1960 and 2030 and piControl and scaled by corresponding global-mean SST increase (i.e. global warming, GW) between the two periods in each model. Stippling indicates statistical significance above the 95% level based on a two-tailed Student's t-test. **b** Inter-model relationship between zonal SST warming contrast and E-minus-C rainfall variability change.

The zonal warming contrast is defined as SST warming difference between the eastern (170°W–100°W, 5°N–5°S) and western (120°E–170°W, 5°N–5°S) equatorial Pacific (dashed boxes in **a**). EP (CP)-ENSO rainfall is diagnosed at centres by regression of anomalous rainfall onto E-index (C-index) over the whole period in each model (see text for details). **c** Inter-model relationship between E-minus-C rainfall variability change and E-minus-C signal difference. The linear fits (solid lines) are displayed in **b**, **c** together with correlation coefficient R and corresponding p-value. Results shown here are based on the RCP85/SSP585 scenario for a future warming climate, focusing on the 60–80-year window.

Alpha in amplitude) systematically generate a larger east-west SST warming differential (Fig. 5c).

Thus, there is a positive feedback loop in which a larger east-west SST warming differential promotes an eastward shift of atmosphere convection toward the eastern equatorial Pacific, where through the non-linear Bjerknes feedback, EP-ENSO SST variability increases more than CP-ENSO, in turn enhancing the zonal SST warming differential. The consequence is a greater E-minus-C signal contrast in SST variability change and a greater ToE contrast between EP-ENSO and CP-ENSO under greenhouse warming.

## Discussion

We find that increased EP-ENSO SST variability emerges more than a decade earlier than that of CP-ENSO and with a higher probability. The earlier EP-ENSO ToE is mainly attributed to a stronger EP-ENSO SST change signal because of an eastward shift and intensification of EP-ENSO rainfall response to a faster SST warming in the eastern equatorial Pacific than the surrounding regions. The faster warming is further amplified by the non-linear Bjerknes feedback, which boosts the

increase in EP-ENSO wind and hence SST variability, ultimately amplifying the ToE contrast between EP-ENSO and CP-ENSO. Under an unmitigated high emission scenario, the EP-ENSO ToE is estimated to be around 2030 ± 6 aggregated over 63% of the 68 climate models using 60–80-year-long records to diagnose noise and signal; using CMIP6 only, 70% (26 out of 37) models show a ToE before the end of the 21st century, yielding a mean ToE of 2026.

Climate models suffer from persistent biases in their simulation of the mean equatorial climate and ENSO dynamics, raising a question as to whether our findings are affected in any way. Although some models still simulate a too-cold climatological Pacific cold tongue, we find no statistically significant (less than the 85% confidence level) inter-model relationship between the cold tongue bias and signal of SST variability change for either EP- or CP-ENSO regime (Supplementary Fig. 10a, b). In addition, models generally underestimate ENSO rainfall response to SST (Supplementary Fig. 10c, d). Inter-model relationship shows that compared to CP-ENSO, the underestimated EP-ENSO rainfall sensitivity to E-index appears to have a stronger impact on E-index variability change, but the underestimate does not show a systematic influence

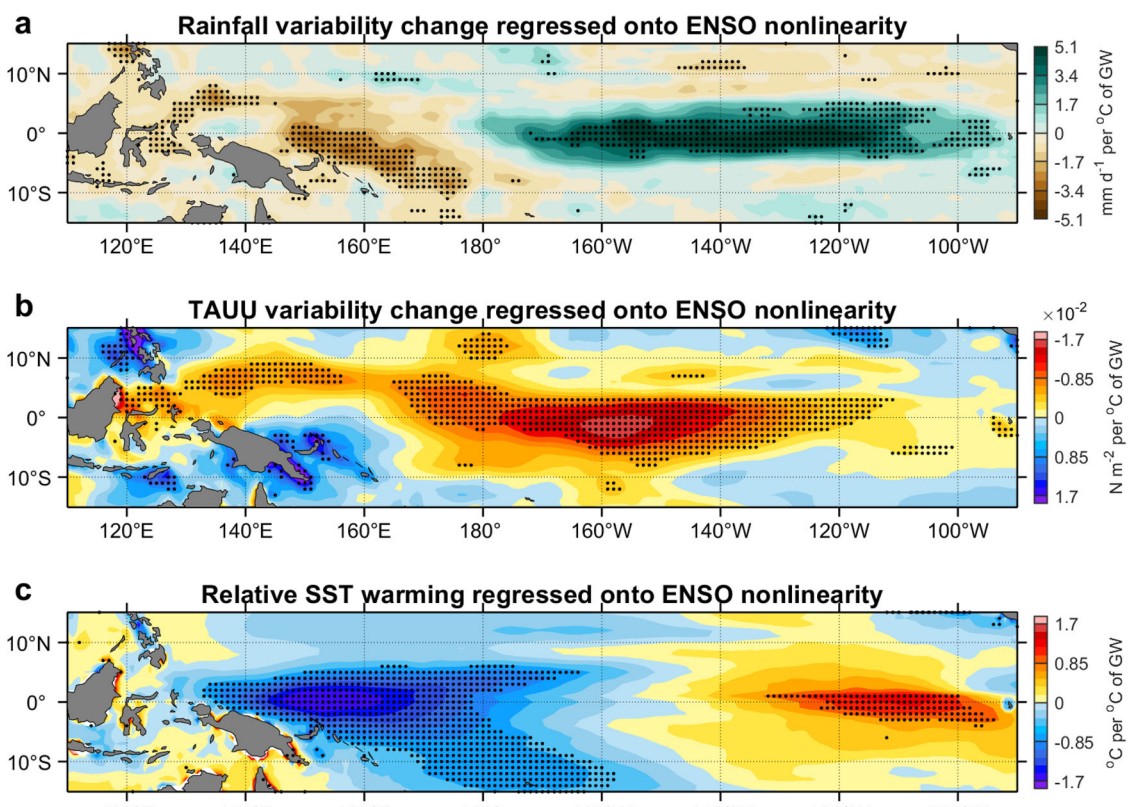

**Fig. 5 | Stronger eastern-Pacific El Niño-Southern Oscillation (EP-ENSO) change signal boosted by stronger non-linear Bjerknes feedback.** Inter-model regression of **a** grid-point rainfall variability change (1960–2030 minus piControl, mm d$^{-1}$ per °C of global warming), **b** grid-point zonal wind stress (TAUU) variability change (1960–2030 minus piControl, N m$^{-2}$ per °C of global warming) and **c** relative sea surface temperature (SST) warming (1960–2030 minus piControl, °C per °C of global warming) onto strength of ENSO non-linearity (measured by amplitude of Alpha over the whole period; see Methods). Black stippling in **a**–**c** indicates statistical significance above the 90% confidence level based on a two-tailed Student's $t$-test. Results shown here are based on the RCP85/SSP585 scenario for a future warming climate.

that is statistically significant. Furthermore, models tend to simulate overly weak ENSO non-linear feedback processes (Supplementary Fig. 1f). However, we find that our conclusion is insensitive to model selection based on simulation of the ENSO non-linear dynamics, although the exact values of ToE and associated uncertainty range may change slightly (Supplementary Fig. 11). If models were to simulate the non-linear feedback processes realistically, the signal of increased E-index variability would be stronger, and the probability of detecting its change would be higher. Finally, we note that the inter-model consistency for ToE of CP-ENSO is less than that of EP-ENSO, potentially due to a smaller SNR of CP-ENSO, indicating that the ToE estimate for CP-ENSO carries a larger uncertainty.

By assessing the ToE of EP-ENSO and CP-ENSO separately and taking into account of availability of some 70 years of reliable observations, we show that ToE of EP-ENSO emerges some four decades earlier than previously suggested[26]. Although detection would always be challenging, especially in the presence of unpredictable internal climate variability, and in a single realization of the real world[54–56], our results highlight that intensified ENSO SST variability under greenhouse warming is more likely to emerge and be detectable in the eastern than in the central equatorial Pacific.

## Methods

### Reanalyses and model outputs
We use three SST reanalysis products, covering a period from 1870 to 2019, to characterise observed ENSO (Supplementary Figs. 1 and 2). These products are: HadISST v1.1 (Hadley Centre Sea Ice and Sea dataset version 1.1)[57], ERSST v3b (Extended Reconstructed Sea Surface Temperature version 3b)[58] and COBE-SST2 (Centennial in situ Observation-Based Estimation Sea Surface Temperature version 2)[59].

For each dataset, SST anomalies are constructed with reference to the monthly climatology of the full period of 1870–2019, and then quadratically detrended.

We take outputs from 31 CMIP5 and 37 CMIP6 coupled global climate models[37,38] based on data availability (Supplementary Table 1). Each model contains a multi-century (>300 years) pre-industrial control (piControl) experiment, a historical run starting from piControl in the mid-19th century and forced with historical forcings up to 2005 for CMIP5 or 2014 for CMIP6, and thereafter future greenhouse gases under the RCP scenario for CMIP5 or the equivalent SSP scenario for CMIP6, with most models sharing a common integration period of 1850–2099.

For each model, SST anomalies referenced to the monthly piControl climatology are first detrended quadratically, separately for the piControl and "historical-plus-future" periods, and then combined to form a concatenated SST anomaly field, to depict ENSO SST evolution in the whole period spanning from piControl to future warming climates. We also use data from 3-D ocean temperature, surface zonal wind stress (TAUU) and rainfall to explore possible mechanisms controlling ToE of the EP- and CP-ENSO regimes. Anomalies of these variables, wherever needed, are calculated in the same way as that of SST. Using different periods of climatology to calculate anomaly does not alter our results. We mainly focus on the RCP85/SSP585 scenario to examine ToE and associated mechanisms and test their sensitivity to various emission scenarios (Supplementary Table 2).

### E and C indices
To depict the two ENSO regimes, at least two indices are needed. As in previous studies[10,23], we adopt a process-based approach to objectively locate EP- and CP-ENSO SST anomaly centres that are unique to each

model. Specifically, these are obtained from a combination of the first two EOF modes of monthly SST anomalies in an equatorial domain (15°S–15°N, 140°E–80°W), each described by a spatial pattern and a principal component (PC) time series normalised to have a unit standard deviation. The EOF1 features a classical El Niño pattern, with anomalous warming broadly distributed in the equatorial central and eastern-Pacific (Supplementary Fig. 1a, c). The EOF2 is characterised by an anomalous cooling in the far east but warming in the central-western equatorial Pacific (Supplementary Fig. 1b, d). The linear combination of EOF1 and EOF2 represents EP- and CP-ENSO events, referred to as E-index defined as $(PC1 − PC2)/\sqrt{2}$ and C-index defined as $(PC1 + PC2)/\sqrt{2}$, such that their associated maximum SST anomalies are in the equatorial eastern and central-Pacific, respectively. We apply this EOF approach to both observations and the concatenated SST anomalies in each CMIP model, to obtain C-index and E-index in the observed period of 1870–2019 and over the simulated period spanning from pre-industrial to a future warming climate.

### Observed and simulated ENSO non-linear dynamics

ENSO is intrinsically non-linear, as manifested by positive and negative skewness of the E-index and C-index, respectively[23] (Supplementary Table 1). The fundamental dynamics relates to a non-linear Bjerknes feedback[35,39,60], where only after substantial warming surpasses a threshold and rare local deep atmospheric convection is triggered, do zonal winds respond to additional warming in the eastern equatorial Pacific. The non-linear zonal wind response leads to further SST warming, resulting in a strong EP El Niño. Strong westerly wind anomalies at the peak of strong EP El Niño tend to induce a large upper-ocean heat discharge and anomalous thermocline shallowing[41], in turn favouring subsequent SST cooling in the CP and thus a strong CP La Niña[22]. As such, the non-linear Bjerknes feedback serves as a bridge linking the two opposite skewed ENSO regimes[42]. The non-linear feedback is, however, distinctively weaker in the CP region where background SST is higher and atmospheric convection occurs more frequently.

The strength of the non-linear Bjerknes feedback is reflected by the amplitude of $|\alpha|$ in a non-linear relationship between the two PCs[23,36] (Supplementary Fig. 1e):

$$PC2(t) = \alpha[PC1(t)]^2 + \beta PC1(t) + \gamma \qquad (1)$$

where $t$ is time, $\alpha$, $\beta$ and $\gamma$ are the non-linear coefficients, linear coefficient and constant of the quadratic function, respectively. Models with a greater $|\alpha|$ systematically produce a greater amplitude of E-index positive skewness and C-index negative skewness (Supplementary Fig. 1f). Therefore, $|\alpha|$ is a simple yet physically sound metric for quantifying ENSO non-linearity. We calculate $|\alpha|$ based on the first two PCs of monthly SST anomalies in both observations and CMIP models.

Aggregated from the reanalyses, the observed amplitude of $|\alpha|$ is around 0.25, larger than a majority of models and their average (0.14) (Supplementary Fig. 1f), consistent with previous findings that many models struggle to realistically simulate the two ENSO types[23]. A stronger $|\alpha|$ means a higher level of non-linearity, more differentiated centres of the two ENSO regimes, and therefore stronger non-linear Bjerknes feedback at the play of the ENSO system[5,23]. Despite still being underestimated, the non-linear ENSO feedback ($|\alpha|$) is better represented in CMIP6 than CMIP5 models, likely contributing to a stronger inter-model consensus on the projected increase of ENSO SST variability in CMIP6 than CMIP5[5].

### ToE of ENSO SST variability change

Among various approaches[27–30], we adopt a "signal-to-noise" method to quantify ToE of ENSO SST variability change. Taking EP-ENSO as an example, the time series of the evolution of its variability is first diagnosed in sliding windows of a given time length (in integer years),

moving every year (12 months) forward from the start of piControl to the end of the 21st century, based on the normalised monthly E-index. Then the piControl mean of the windowed variability is removed from the whole time series to produce the variability change relative to its mean pre-industrial level. From the new time series, noise is defined as one standard deviation (1.0 s.d.) of the time series over the piControl period, as any unforced internal variability can be viewed as background "noise" to warming-induced ENSO SST variability change in the 20th and 21st century, which we refer to as a signal. Finally, ToE is defined as the time when (in integer year) the windowed E-index variability change, relative to the piControl mean, first exceeds a threshold of two times of noise (i.e. Signal-to-Noise-Ratio, SNR > 2.0) and remains above the threshold thereafter (Fig. 1b, d). ToE is recorded as the last year of the sliding window of emergence. The same procedures are applied to C-index.

We vary the length of sliding windows (in integer years) over which SST variability is diagnosed to obtain ToE as a function of window length. For each given window, ToE is first calculated for each model and then averaged across all models that show emerging signals before 2100. Using a multi-model median yield an essentially similar result (Fig. 2c, d). The threshold of signal exceedance over noise is chosen at 2.0 based on the large consensus in existing literature[29] and because it represents a 95% confidence of signal emergence. Sensitivity to a more stringent threshold value is tested (Supplementary Table 2). Overall, this chosen threshold (2.0) makes it a rather high (thus conservative) estimate of the noise envelope (Fig. 1b, d). We have also tested that the integration period of piControl does not materially affect the noise level provided it is long enough (e.g. >300 years) to ensure a large sample size for noise estimation, as is the case in the present study.

### Bootstrap test

Given the limited number of CMIP models, a Bootstrap method[61] is used to provide a statistical context of significance. Specifically, in a given length of the time window, all models that show ToE before 2100 are resampled randomly to construct 10,000 realisations of a multi-model mean ToE. In this resampling process, any model is allowed to be selected again. The standard deviation of the 10,000 realisations is then calculated as the uncertainty range of the multi-model mean estimate. Further, if the difference of the multi-model mean value between two groups (e.g. E-index vs C-index) is greater than the sum of the two separate 10,000-realization standard deviation values, then the difference is considered statistically significant above the 95% confidence level.

## Data availability

Data related to the paper can be downloaded from the following: HadISST v1.1, https://www.metoffice.gov.uk/hadobs/hadisst/; ERSST v3b, https://psl.noaa.gov/data/gridded/data.noaa.ersst.v3.html; COBE SST2, https://psl.noaa.gov/data/gridded/data.cobe2.html; CMIP5 datasets, https://esgf-node.llnl.gov/projects/cmip5/; CMIP6 datasets, https://esgf-node.llnl.gov/projects/cmip6/.

## Code availability

Codes for calculating EOF and the parameter $|\alpha|$ can be downloaded from: https://drive.google.com/open?id=1d2R8wKpFNW-vMIfoJsbqIGPIBd9Z_8rj. Other codes are available upon request.

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

## Acknowledgements

This study is supported by the National Natural Science Foundation of China (NSFC) project (42206209), the Strategic Priority Research Program of Chinese Academy of Sciences (XDB40030000), and the National Key R&D Program of China (2018YFA0605700). T.G. is supported by China National Postdoctoral Program for Innovative Talents (BX20220279). W.C. is supported by the Centre for Southern Hemisphere Oceans Research, a joint research centre between QNLM and CSIRO. W.C., G.W. and A.S. are also supported by the Earth Systems and Climate Change Hub of the Australian Government's National Environmental Science Program. PMEL contribution no. 5305.

## Author contributions

T.G. and W.C. designed the study and wrote the initial manuscript in discussion with L.W. T.G. performed analysis and generated all figures. A.S. and G.W. contributed to finalising the paper. Z.J., B.G., Y.Y., S.L., S.W. and Z.C. contributed to interpreting results and discussion of the associated dynamics. M.M. contributed to improvement of this paper.

## Competing interests

The authors declare no competing interests.
