## [Peer Review File · Nature Communications]

Emergence of changing Central-Pacific and Eastern-Pacific El Niño-Southern Oscillation in a warming climateREVIEWER COMMENTS

Reviewer #1 (Remarks to the Author):

General Comments:

This paper focus on the emergence of changing Central-Pacific and Eastern-Pacific ENSO in a warming climate. As the two types of ENSO can exert contrasting climatic impacts, it is worth exploring the diverse response of each ENSO regime to greenhouse warming. Overall, the results of this study is interesting, but the manuscript requires further improvements to address a few major comments.

Major Concerns:

1. As we all known, the current CGCMs are imperfect to depict the evolution of two ENSO regimes. Some CMIP models even have no ability to represent the diversity of ENSO. In this study, the authors collected abundant CMIP models, but did not consider their actual ability to simulate the diversity of ENSO. Maybe the MME from the models that can reasonably represent the two ENSO regimes is more persuasive to investigate this topic.
2. Although the model has the basic ability to represent the diversity of ENSO, there are still some model errors that can contaminate the response of the each ENSO regime to greenhouse warming. The analysis of these study should consider the effect of the model errors on your results. The correction of the general model bias is least to do before further study.
3. The rainfall response to SST warming is one of major uncertainties in the climate models. So the model ability to represent this process should be evaluated, and the influence of corresponding model bias on the results need to be considered.
4. The CP ENSO mainly occurs after 1960-1970. Whether the results will change when selecting the mean before this period rather than the piControl mean to calculate the ToE of CP ENSO?

Reviewer #2 (Remarks to the Author):

Review on “Emergence of changing Central-Pacific and Eastern-Pacific El Niño-Southern Oscillation in a warming climate” by Geng et al.

The study highlights the timing of the emergence of Eastern Pacific (EP) and Central-Pacific (CP) ENSO events in a warming climate. They found that increased EP-ENSO SST variability emerges more than a decade earlier than that of CP-ENSO and with a higher probability. The earlier EP-ENSO ToE is mainly attributed to a stronger EP-ENSO SST change signal because of an eastward shift and intensification of EP- ENSO rainfall response to a faster SST warming in the eastern equatorial Pacific than in the surrounding regions. Overall, the study is very interesting and the analysis is consistent. The presentation of the idea is also very nice, and progressively increases the depth of the concept. Therefore, I would like to recommend publication after some revisions and clarifying the following queries.

Line 137-141. The authors have considered the length of the time window in the range ~60-80 years however, they have not clearly stated why they have chosen only this particular time window. In the following section, it is mentioned that the availability of reliable observed SST is available for 70 years. It is confusing that the time of window here is based on the simulations. It would be nice if the authors could mention why they have considered this time range?

Line 160-164. In general, CMIP6 models are containing strong mean state biases, especially in the tropical Pacific cold-tongue region. I wonder what is the role of these mean state biases in the emergence of EP ENSO events. Authors may add some discussion by keeping this aspect as well.

Line 83. Typo error in C index equation.

Reviewer #3 (Remarks to the Author):

In the current manuscript the authors address a very significant and strongly debatable issue concerning ENSO modification under global warming conditions. They present the original noteworthy results highlighting the different response of two ENSO regimes to the greenhouse warming. The evidence of earlier than previously supposed SST variability increase has undoubted practical relevance allowing to develop the appropriate response strategies. I have appreciated the very thorough treatment of data and sophisticated methods of the analysis allowing to obtain the reliable conclusions.

I recommend this manuscript for publication after minor revisions.

I have two main questions/comments

1. I find some inconsistencies in the terminology used. Considering the heat budget of upper mixed layer during ENSO the following terminology is commonly used

$$\partial T'/\partial t \approx -u'(\partial \bar{T})/\partial x - w'(\partial \bar{T})/\partial z - \bar{w}'\partial T'/\partial x$$

where the first term in the right-hand side represents zonal advective feedback, the second term represents Ekman pumping-induced anomalous upwelling feedback, and the third term represents Bjerknes thermocline feedback.

In your study you discuss the nonlinear Bjerknes feedback that consists in interaction between SST, zonal wind and convection anomalies, but not thermocline displacement. It seems to me that the reader may confuse these two types of Bjerknes feedbacks. I suggest to describe more in details the feedbacks involved in the ENSO mechanism and precise what type of feedback is involved in the ENSO response to greenhouse warming.

2. Recent investigation have demonstrated that two ENSO regimes induce different response in mid to high latitudes of both hemispheres, with strong interhemispheric difference, as well as the different response in troposphere and stratosphere. Moreover, the significant modification of this response in future climate was evidenced. This fact may be mentioned in the introduction.

The minor comments are listed below

Lines 110. What do you mean as "SST variability"? RMS of SST timeseries or something else? It should be clarified in the text.

Lines 126-127. "Varying the length of sliding windows yields ToE as a function of length of period in which the signal and noise are diagnosed." Did you diagnose the relationship between the length of sliding window and ToE? How the length of period in which the signal and noise are diagnosed affect ToE? Are your conclusions sensible to the changes of period length? If you discuss these results later it is recommended to move this phrase toward the discussion to not provoke additional questions.

Line 136. The figure capture of Supplementary Fig.2 is not complete: it is unclear what mean the different symbols (triangles, squares etc) of different colors.

Lines 137-138. "the increase in ENSO SST variability over a multi-centennial period does not guarantee its emergence before 2100". Emergence of what? Of the increase of SST variability? If the variability is calculated over some period how you define its emergence?

Lines 137-144 "However, the increase in ENSO SST variability over a multi-centennial period does not guarantee its emergence before 2100, since the ToE of each ENSO regime depends on the length of time over which background SST noise and signal are diagnosed (Supplementary Fig.3). For example, in the range of 60-80 years, only 63.2% of models show a ToE of E-index and 141

33.8% of a ToE of C-index before 2100. For a given model, the longer the time window, the lower the noise level of unforced natural variability is compared to the warming-induced SST change signal and a larger SNR (Supplementary Fig. 4), thus facilitating an earlier emergence of the signal and a higher probability of the emergence (Fig. 2a, b)."

I do not fully understand this discussion. If the ToE is strongly influenced by the length of the period, how valuable are your findings? They are applicable only for the specific length of period (for example 60-80 years) but are wrong for the shorter period?

Lines 313-314. "We find that increased EP-ENSO SST variability emerges more than a decade earlier than that of CP-ENSO and with a higher probability" See the comment above: "increasedvariability emerges earlier". It is contradictory as the variability is calculated for the period and therefore can't emerge earlier. You may do this conclusion if you estimate the variability for separate periods and identify the date when the variability starts to increase with any changes before. But, as I understand, you do not apply such approach. Need to rephrase.

Point-by-Point Response to Reviewers (our response in blue)

We thank all three reviewers for their positive and helpful comments.

Response to Reviewer #1

General Comments:

This paper focus on the emergence of changing Central-Pacific and Eastern-Pacific ENSO in a warming climate. As the two types of ENSO can exert contrasting climatic impacts, it is worth exploring the diverse response of each ENSO regime to greenhouse warming. Overall, the results of this study is interesting, but the manuscript requires further improvements to address a few major comments.

Thank you for your time and effort in evaluating our paper!

Major Concerns:

1. As we all known, the current CGCMs are imperfect to depict the evolution of two ENSO regimes. Some CMIP models even have no ability to represent the diversity of ENSO. In this study, the authors collected abundant CMIP models, but did not consider their actual ability to simulate the diversity of ENSO. Maybe the MME from the models that can reasonably represent the two ENSO regimes is more persuasive to investigate this topic.

Thanks for the suggestion. We have included discussion of sensitivity to model realism in simulating ENSO diversity. We measure model ability to simulate ENSO diversity by *Alpha* (Supplementary Fig. 1e, f). A stronger (more negative) *Alpha* means larger skewness in amplitude of both E-index and C-index and more differentiated centres of the two ENSO regimes (Ref.5; Ref.23). We have selected 42 (out of 68) models with a value of *Alpha* that is at least one third of the observed *Alpha* (-0.25) to test sensitivity of our results to model selection.

As shown in **Fig. R1** below, our key results hold, including ToE decreasing with increasing lengths of time window and EP-ENSO emerging earlier than CP-ENSO. For example, in the range of 60-80 years, 78.57% (33 out of 42) of all models generate a ToE of EP-ENSO with a MMM value of ~2026, whereas only 38.10% (16 out of 42) of models generate a ToE of CP-ENSO with a MMM value of ~2035. Using other thresholds (0 or 1/2 of observed amplitude) of *Alpha* to select models yields similar results. Therefore, our results are qualitatively robust.

We have included this analysis in revised discussion section (Lines 335-349).

Fig. R1 | Same as Fig. 2, but for 42 selected models of which *Alpha* is at least one third of the observed value.

2. Although the model has the basic ability to represent the diversity of ENSO, there are still some model errors that can contaminate the response of the each ENSO regime to greenhouse warming. The analysis of these study should consider the effect of the model errors on your results. The correction of the general model bias is least to do before further study.

Great suggestion. In the revised version, we have assessed potential impacts from two common biases.

Although most models still simulate a too-cold climatological cold tongue in the equatorial Pacific, there is no significant (less than the 85% confidence level) inter-model relationship between the intensity of the cold tongue bias and signal of ENSO SST variability change for either EP- or CP-ENSO regime (**Fig. R2 a, b**), suggesting that the mean SST bias does not systematically affect our results.

(The following relates to your major comment #3). We have also evaluated model ability to represent rainfall response to SST. Models tend to underestimate rainfall sensitivity to SST for both EP- and CP-ENSO regimes (compare vertical solid and dashed lines in Fig.R2 c, d), which is potentially linked with a too-cold climatological cold tongue, inconducive to atmospheric convection. An inter-model relationship suggests that compared to CP-ENSO, rainfall sensitivity to EP-ENSO SST in the present climate has a stronger impact on projected E-index variability change, but it only accounts for ~10% ($R=0.33$) of inter-model variation of the latter. Noise is less influenced by this bias. The underestimation of EP rainfall sensitivity to E-index further suggests that if this bias were to be corrected, models would produce a larger increase in EP-ENSO SST variability and thus an earlier ToE of the EP-ENSO SST variability change than CP-ENSO ToE. That is, our conclusion of earlier EP-ENSO ToE than CP-ENSO ToE would be even firmer.

We have included Fig.R2 as Supplementary Fig. 10 and added a separate paragraph about model biases in the revised discussion section (Lines 335-349).

Fig. R2 (new Supplementary Fig. 10) | Influence of model bias on signal of ENSO SST variability change. **a**, Inter-model relationship between cold tongue bias ($^{\circ}\text{C}$), measured by model-observation difference in climatological SST over the equatorial Pacific region (160°E - 100°W , 2°S - 2°N) in 1900-1999, and 60-80-year E-index signal (s.d.). The vertical dashed and solid lines indicate observed (obs.) and multi-model mean (MMM.) values of the bias, respectively. The linear fit (red solid line) is displayed together with correlation coefficient R and corresponding p value. **b**, As in **a**, but for C-index. **c**, **d**, Same as **a**, **b**, respectively, but for inter-model relationship between rainfall sensitivity to SST and 60-80-year ENSO SST variability signal. The rainfall sensitivity is calculated for each individual model as the regression coefficient of normalized rainfall anomalies (5°S - 5°N average) onto E-index or C-index at corresponding maximum response centres in 1900-1999. Observed rainfall is from GPCP⁶³ and SST is from HadISST⁵⁸. Results shown here are based on the RCP85/SSP585 scenario for a future warming climate, focusing on the 60-80-year window. The 47 models are used that show a ToE of either E-index or C-index before 2100 using the 60-80-year windows.

3. The rainfall response to SST warming is one of major uncertainties in the climate models. So the model ability to represent this process should be evaluated, and the influence of corresponding model bias on the results need to be considered.

Please see our response above.

4. The CP ENSO mainly occurs after 1960-1970. Whether the results will change when selecting the mean before this period rather than the piControl mean to calculate the ToE of CP ENSO?

We have repeated our analysis using 1850-1950 mean to calculate ToE and found that our results little changed (**Fig. R3**). We have added this aspect in the Methods section (Line 520-521)

Fig. R3 | Same as Fig. 2a, b, respectively, but using 1850-1950 mean to calculate ToE.

Response to Reviewer #2

Review on “Emergence of changing Central-Pacific and Eastern-Pacific El Niño-Southern Oscillation in a warming climate” by Geng et al.

The study highlights the timing of the emergence of Eastern Pacific (EP) and Central-Pacific (CP) ENSO events in a warming climate. They found that increased EP-ENSO SST variability emerges more than a decade earlier than that of CP-ENSO and with a higher probability. The earlier EP-ENSO ToE is mainly attributed to a stronger EP-ENSO SST change signal because of an eastward shift and intensification of EP- ENSO rainfall response to a faster SST warming in the eastern equatorial Pacific than in the surrounding regions. Overall, the study is very interesting and the analysis is consistent. The presentation of the idea is also very nice, and progressively increases the depth of the concept. Therefore, I would like to recommend publication after some revisions and clarifying the following queries.

Thank you for your positive and helpful comments!

Line 137-141. The authors have considered the length of the time window in the range ~60-80 years however, they have not clearly stated why they have chosen only this particular time window. In the following section, it is mentioned that the availability of reliable observed SST is available for 70 years. It is confusing that the time of window here is based on the simulations. It would be nice if the authors could mention why they have considered this time range?

Thank you for the question. We chose this particular time range (60-80 years) for two reasons.

First, a feature of the results is that the dependence upon time length exhibits a saturation behaviour (Fig. 2a, b), as internal variability does not decrease further with longer time length and a further increase in time length changes little the warming-induced signal. The behaviour indicates that a time window longer than approximately 60-80 years would hardly further benefit detection of ENSO variability change.

Second, as we stated, only ~70 years of relatively reliable SST data are available so far because observations before 1950 carry a large uncertainty. Therefore, given the saturation behaviour and to highlight relevance to observed ENSO change detection, we focus on the time length of 60-80 years.

We have mentioned these reasons in the revised manuscript and rearranged corresponding paragraphs to make the paper flow more logically (Lines143-160).

Line 160-164. In general, CMIP6 models are containing strong mean state biases, especially in the tropical Pacific cold-tongue region. I wonder what is the role of these mean state biases in the emergence of EP ENSO events. Authors may add some discussion by keeping this aspect as well.

We have assessed potential impacts from the cold tongue bias on our results. An inter-model relationship suggests that models with a stronger cold tongue bias tend to simulate a weaker signal of E-index variability change and thus likely a later ToE of EP-ENSO (Fig. R4), but the tendency is NOT statistically significant ($p=0.11$).

We have added a separate paragraph about model biases in the revised discussion section (Lines 335-349).

Fig. R4 | Influence of cold tongue bias on ToE of EP-ENSO. Inter-model relationship between cold tongue bias ($^{\circ}\text{C}$), measured by model-observation difference in climatological SST over the equatorial Pacific region (160°E - 100°W , 2°S - 2°N) in 1900-1999, and 60-80-year ToE of E-index. The vertical dashed and solid lines indicate observed (obs.) and multi-model mean (MMM.) values of the bias, respectively. The linear fit (red solid line) is displayed together with the coefficient R and the corresponding p value. There is no systematic influence from the cold tongue bias.

Line 83. Typo error in C index equation.

Corrected. Thanks.

Response to Reviewer #3

In the current manuscript the authors address a very significant and strongly debatable issue concerning ENSO modification under global warming conditions. They present the original noteworthy results highlighting the different response of two ENSO regimes to the greenhouse warming. The evidence of earlier than previously supposed SST variability increase has undoubted practical relevance allowing to develop the appropriate response strategies. I have appreciated the very thorough treatment of data and sophisticated methods of the analysis allowing to obtain the reliable conclusions.

I recommend this manuscript for publication after minor revisions.

Thank you for your positive and helpful comments!

I have two main questions/comments

1. I find some inconsistencies in the terminology used. Considering the heat budget of upper mixed layer during ENSO the following terminology is commonly used $\partial T'/\partial t \approx -u'(\partial \bar{T})/\partial x - w'(\partial \bar{T})/\partial z - \bar{w} \partial T'/\partial x$ where the first term in the right-hand side represents zonal advective feedback, the second term represents Ekman pumping-induced anomalous upwelling feedback, and the third term represents Bjerknes thermocline feedback. In your study you discuss the nonlinear Bjerknes feedback that consists in interaction between SST, zonal wind and convection anomalies, but not thermocline displacement. It seems to me that the reader may confuse these two types of Bjerknes feedbacks. I suggest to describe more in details the feedbacks involved in the ENSO mechanism and precise what type of feedback is involved in the ENSO response to greenhouse warming.

Thank you for the suggestion. The nonlinear Bjerknes feedback we discussed does include interaction between SST, zonal wind, convection anomalies, **and thermocline displacement, or precisely, the thermocline feedback**. It is the nonlinear dependence of atmospheric convection on EP SST that disproportionately influences the zonal wind response, and subsequent interactions between zonal wind, SST, thermocline, and convection anomalies.

We have made this clear in the revised manuscript (Lines 102-105).

2. Recent investigation have demonstrated that two ENSO regimes induce different response in mid to high latitudes of both hemispheres, with strong interhemispheric difference, as well as the different response in troposphere and stratosphere. Moreover, the significant modification of this response in future climate was evidenced. This fact may be mentioned in the introduction.

Done. We have added this in the introduction (Lines 49-51).

The minor comments are listed below

Lines 110. What do you mean as “SST variability”? RMS of SST timeseries or something else? It should be clarified in the text.

The “SST variability” refers to standard deviation of SST time series. This has been clarified in the revised manuscript (Line 113).

Lines 126-127. “Varying the length of sliding windows yields ToE as a function of length of period in which the signal and noise are diagnosed.” Did you diagnose the relationship between the length of sliding window and ToE? How the length of period in which the signal and noise are diagnosed affect ToE? Are your conclusions sensible to the changes of period length? If you discuss these results later it is recommended to move this phrase toward the discussion to not provoke additional questions.

Yes, we discussed all these aspects in the next section entitled “Earlier emergence of EP-ENSO change than CP-ENSO”. Following your suggestion, we have removed the sentence to make the paper flow better. As such, the questions you asked are addressed in our response to your comments below.

Line 136. The figure capture of Supplementary Fig.2 is not complete: it is unclear what mean the different symbols (triangles, squares etc) of different colors.

Thank you for picking this up. We have added corresponding description in the caption of Supplementary Fig.2 (Line 742-743).

Lines 137-138. “the increase in ENSO SST variability over a multi-centennial period does not guarantee its emergence before 2100”. Emergence of what? Of the increase of SST variability? If the variability is calculated over some period how you define its emergence?

We mean emergence of “increased SST variability induced by greenhouse warming” out of internal variability. The amplitude range of internal variability depends on time scale of variability considered. For example, the amplitude of *decadal* variability is usually greater than the amplitude of *multi-decadal* variability.

We have time series of ENSO SST variations in terms of *E-index* and *C-index*, respectively, each of which includes variability of all time scales. To calculate E-index variability on a time scale of 30 years, we compute standard deviation in the first 30-year time window. We then move the 30-year window forward every year to calculate standard deviation each time, covering from the piControl to the end of the 21st century. In this way we obtain a time series of fluctuating values of standard deviation covering the piControl period to the end of the 21st century (**Fig. R5a, an example from GFDL-ESM4**).

We define “noise” as the 95-percentile value of the time series of fluctuating values of standard deviation of E-index *over the piControl period only* in which there is no climate change forcing (shaded period in **Fig. 5a**).

ToE is defined as when the sliding SST variability *from 1850 to the end of the 21st century* exceeds the 95-percentile threshold value of the sliding standard deviation values *over the piControl period* (i.e., Signal-to-Noise Ratio, SNR>2.0) and remains above the threshold

thereafter. For the 30-year time window length, the ToE for E-index is 2064 (**Fig. R5a, red vertical time**).

Therefore, an exact ToE depends on time length of sliding windows used for diagnosis because the time window length influences noise (i.e., the 95-percentile threshold) for ENSO SST variability change. The longer the time window length, the smaller the noise threshold. Further, the longer the time window length, the greater the climate change signal. Please see **Fig. R5** below for an illustration of the dependence.

We have incorporated the above discussion in the revised manuscript (Line 119-126).

Fig. R5 | Same as Fig. 1b, but for evolution of E-index variability (black solid line) and ToE (red star) diagnosed in different lengths (30, 50, 80 years) of time window from GFDL-ESM4, illustrating the influence of length of time window on estimating noise, signal, and ToE. **The longer the time window length, the smaller the noise threshold (vertical range in the shaded period).**

Lines 137-144 “However, the increase in ENSO SST variability over a multi-centennial period does not guarantee its emergence before 2100, since the ToE of each ENSO regime depends on the length of time over which background SST noise and signal are diagnosed (Supplementary Fig.3). For example, in the range of 60-80 years, only 63.2% of models show a ToE of E-index and 141 33.8% of a ToE of C-index before 2100. For a given model, the longer the time window, the lower the noise level of unforced natural variability is compared to the warming-induced SST change signal and a larger SNR (Supplementary Fig. 4), thus facilitating an earlier emergence of the signal and a higher probability of the emergence (Fig. 2a, b).”

I do not fully understand this discussion. If the ToE is strongly influenced by the length of the period, how valuable are your findings? They are applicable only for the specific length of period (for example 60-80 years) but are wrong for the shorter period?

Our findings are valuable in that we demonstrate time length of period in which signal and noise are diagnosed affects ToE of ENSO SST variability change. It illustrates that a longer period of reliable observations helps detection of ENSO change under greenhouse warming.

For example, compared to 60-80 years, using 30-year time length may not be as helpful for detection of ENSO SST variability change because the relatively large internal variability (“noise”) can more easily mask the greenhouse-warming-induced ENSO change signal, thereby leading to a later ToE, which can be after 2100 (Fig.2 a, b; **Fig. R5**).

If we shorten the time length further, for example, to 10 years, the chance of detecting a change in ENSO SST variability is even lower.

We have highlighted the relevance for observed ENSO change in the revision (Lines 153-160).

Lines 313-314. “We find that increased EP-ENSO SST variability emerges more than a decade earlier than that of CP-ENSO and with a higher probability” See the comment above: “increasedvariability emerges earlier”. It is contradictory as the variability is calculated for the period and therefore can’t emerge earlier. You may do this conclusion if you estimate the variability for separate periods and identify the date when the variability starts to increase with any changes before. But, as I understand, you do not apply such approach. Need to rephrase.

We should have made this clearer.

We did estimate SST variability for *two separate periods*: unforced piControl period (to determine the “noise” level), and the greenhouse warming period since 1850 to identify the time when the signal moves out of the noise threshold. This process is carried out for E-index and C-index separately. The earlier emergence of EP-ENSO relative to that of CP-ENSO results from their different response to greenhouse warming (“signal”) relative to their respective “noise” level.

We have made these points clearer throughout the paper (see, e.g., Lines 119-126; Lines 143-145). Thank you.

REVIEWERS' COMMENTS

Reviewer #1 (Remarks to the Author):

General Comments:

The science in the manuscript has improved. It is a nicely written paper that is well-organized and logical. There are several minor comments to be addressed before the manuscript is accepted for publication.

Comments:

1. There are still some difference between the result of the whole models and that of the 42 selected models. The uncertainties of the two types of ENSO overlap in the 42 selected models. It means that the earlier ToE of EP ENSO than CP ENSO includes some uncertainties after selecting models. Generally speaking, the result based on the selected models is more convincing. It is better to add Fig. R1 in the Supplementary and present some discussion about the difference between the result of the whole models and that of the selected models.
2. The inter-model consistency for ToE of CP ENSO is less that of EP ENSO. This indicate the ToE of CP ENSO shows some uncertainties. Some related discussion is needed about this point.

Reviewer #2 (Remarks to the Author):

The authors addressed all my previous comments satisfactory. Therefore, I recommend this manuscript for publication.

Reviewer #3 (Remarks to the Author):

The authors have addressed all my comments and presented the detailed and complete answers. They have also made significant revisions concerning the comments of other reviewers that improved the manuscript. I may recommend it for publication

Point-by-Point Response to Reviewers (our response in blue)

We thank all three reviewers for their time and efforts in evaluating our manuscript.

Response to Reviewer #1

General Comments:

The science in the manuscript has improved. It is a nicely written paper that is well-organized and logical. There are several minor comments to be addressed before the manuscript is accepted for publication.

Thank you for your positive and helpful comments!

Comments:

1. There are still some difference between the result of the whole models and that of the 42 selected models. The uncertainties of the two types of ENSO overlap in the 42 selected models. It means that the earlier ToE of EP ENSO than CP ENSO includes some uncertainties after selecting models. Generally speaking, the result based on the selected models is more convincing. It is better to add Fig. R1 in the Supplementary and present some discussion about the difference between the result of the whole models and that of the selected models.

We have now added Fig.R1 as Supplementary Fig. 11 and presented discussion on model selection as suggested. Please see Lines 344-347 in the revision.

2. The inter-model consistency for ToE of CP ENSO is less that of EP ENSO. This indicate the ToE of CP ENSO shows some uncertainties. Some related discussion is needed about this point.

We have mentioned this point as suggested. Please see Lines 349-351 in the revision.

Response to Reviewer #2

The authors addressed all my previous comments satisfactory. Therefore, I recommend this manuscript for publication.

Thank you!

Response to Reviewer #3

The authors have addressed all my comments and presented the detailed and complete answers. They have also made significant revisions concerning the comments of other reviewers that improved the manuscript. I may recommend it for publication

Thank you!